# Some New Generalization of Darbo's Fixed Point Theorem and Its Application on Integral Equations

**Anupam Das** [1,†]**, Bipan Hazarika** [1,2,†] **and Poom Kumam** [3,*,†]

1   Department of Mathematics, Rajiv Gandhi University, Rono Hills, Doimukh 791112, Arunachal Pradesh, India; math.anupam@gmail.com
2   Department of Mathematics, Gauhati University, Guwahati 781014, Assam, India; bh_rgu@yahoo.co.in
3   MUTTFixed Point Research Laboratory, Room SCL 802 Fixed Point Laboratory, Science Laboratory Building, Department of Mathematics, Faculty of Science, King Mongkut's University of Technology Thonburi (KMUTT), 126 Pracha Uthit Rd., Bang Mod, Thung Khru, Bangkok 10140, Thailand
*   Correspondence: poom.kum@kmutt.ac.th
†   These authors contributed equally to this work.

**Abstract:** In this article, we propose some new fixed point theorem involving measure of noncompactness and control function. Further, we prove the existence of a solution of functional integral equations in two variables by using this fixed point theorem in Banach Algebra, and also illustrate the results with the help of an example.

**Keywords:** measure of noncompactness; functional integral equations; Darbo fixed point theorem

## 1. Introduction

Integral equations play a significant role in real-world problems. Fixed point theory and measure of noncompactness are useful tools in solving different types of integral equations which we come across in different real life situations. In solving functional integral equations, Schauder and Darbo's fixed point theorems play a significant role. We refer (see [1–15]) for application of fixed point theorems and measure of noncompactness for solving differential and integral equations.

In this article using the concept of control function and measure of noncompactness we have proved some new fixed point theorems. Further, we have also applied this theorem to study the existence of solution of functional integral equations in Banach algebra and also with the help of an example we have verified our results.

Let $\bar{E}$ be a real Banach space with the norm $\| \cdot \|$. Let $B(a,b)$ be a closed ball in $\bar{E}$ centered at $a$ and with radius $b$. If $X$ is a nonempty subset of $\bar{E}$ then by $\bar{X}$ and Conv $X$ we denote the closure and convex closure of $X$, respectively. Moreover, let $\mathcal{M}_{\bar{E}}$ denote the family of all nonempty and bounded subsets of $\bar{E}$ and $\mathcal{N}_{\bar{E}}$ its subfamily consisting of all relatively compact sets. We denote by $\mathbb{R}$ the set of real numbers and $\mathbb{R}_+ = [0, \infty)$.

The following definition of a measure of noncompactness given in [3].

**Definition 1.** *A function $\mu : \mathcal{M}_{\bar{E}} \to [0, \infty)$ is called a measure of non-compactness in $\bar{E}$ if it satisfies the following conditions:*

*(i)   for all $Y \in \mathcal{M}_{\bar{E}}$, we have $\mu(Y) = 0$ implies that $Y$ is precompact.*
*(ii)  the family ker $\mu = \{Y \in \mathcal{M}_{\bar{E}} : \mu(Y) = 0\}$ is nonempty and ker $\mu \subset \mathcal{N}_{\bar{E}}$.*
*(iii) $Y \subseteq Z \implies \mu(Y) \leq \mu(Z).$*

(iv)  $\mu\left(\bar{Y}\right) = \mu\left(Y\right)$.

(v)  $\mu\left(ConvY\right) = \mu\left(Y\right)$.

(vi)  $\mu\left(\lambda Y + (1-\lambda)Z\right) \leq \lambda\mu\left(Y\right) + (1-\lambda)\mu\left(Z\right)$ for $\lambda \in [0,1]$.

(vii) if $Y_n \in \mathcal{M}_{\bar{E}}$, $Y_n = \bar{Y}_n$, $Y_{n+1} \subset Y_n$ for $n = 1,2,3,\dots$ and $\lim\limits_{n\to\infty} \mu\left(Y_n\right) = 0$ then $\bigcap_{n=1}^{\infty} Y_n \neq \phi$.

The family ker $\mu$ is said to be the *kernel of measure* $\mu$. Observe that the intersection set $Y_\infty$ from (vii) is a member of the family ker $\mu$. In fact, since $\mu(Y_\infty) \leq \mu(Y_n)$ for any $n$, we infer that $\mu(Y_\infty) = 0$. This gives $Y_\infty \in \ker\mu$.

For a bounded subset $S$ of a metric space $X$, the Kuratowski measure of noncompactness is defined as [9]

$$\alpha\left(S\right) = \inf\left\{\delta > 0 : S = \bigcup_{i=1}^{n} S_i, \ \mathrm{diam}\left(S_i\right) \leq \delta \ \text{for } n \in \mathbb{N}\right\},$$

where $\mathrm{diam}\left(S_i\right)$ denotes the diameter of the set $S_i$, that is

$$\mathrm{diam}\left(S_i\right) = \sup\left\{d(x,y) : x,y \in S_i\right\}.$$

The Hausdorff measure of noncompactness for a bounded set $S$ is defined as

$$\chi\left(S\right) = \inf\left\{\epsilon > 0 : S \text{ has finite } \epsilon - \text{net in } X\right\}.$$

**Definition 2** ([3]). *Let X be a nonempty subset of a Banach space $\bar{E}$ and $T : X \to \bar{E}$ is a continuous operator transforming bounded subset of X to bounded ones. We say that T satisfies the Darbo condition with a constant k with respect to measure $\mu$ provided $\mu(TY) \leq k\mu(Y)$ for each $Y \in \mathcal{M}_{\bar{E}}$ such that $Y \subset X$.*

We recall following important theorems:

**Theorem 1** (Shauder [16]). *Let D be a nonempty, closed and convex subset of a Banach space $\bar{E}$. Then every compact, continuous map $T : D \to D$ has at least one fixed point.*

**Theorem 2** (Darbo [10]). *Let Z be a nonempty, bounded, closed and convex subset of a Banach space $\bar{E}$. Let $S : Z \to Z$ be a continuous mapping. Assume that there is a constant $k \in [0,1)$ such that*

$$\mu(SM) \leq k\mu(M), \ M \subseteq Z.$$

*Then S has a fixed point.*

In order to establish our fixed point theorem, we need some of the following related concepts. Khan et al. [17] used a control function which they called an *altering distance function*.

**Definition 3** ([17]). *An altering distance function is a continuous, nondecreasing mapping $\delta : \mathbb{R}_+ \to \mathbb{R}_+$ such that $\delta^{-1}(\{0\}) = \{0\}$.*

**Definition 4.** *We denote $\hat{Z}$ be the class of functions $\eta : \mathbb{R}_+ \times \mathbb{R}_+ \to \mathbb{R}$ satisfying the following conditions:*

(1)   $\eta(0,0) = 0$

(2)   $\eta(t,s) < s - t$ for all $t,s > 0$

(3)   if $\{t_n\}, \{s_n\}$ are sequences in $(0,\infty)$ such that $\lim\limits_{n\to\infty} t_n = t$, $\lim\limits_{n\to\infty} s_n = s > 0$, then $\limsup\limits_{n\to\infty} \eta(t_n, s_n) < s - t$.

For example, let $\psi_1$ and $\psi_2$ be two altering distance functions such that $\psi_1(t) < t \leq \psi_2(t)$ for all $t > 0$. Then $\eta_1(t,s) = \psi_1(s) - \psi_2(t)$ for all $t,s \in \mathbb{R}_+$ is in the class of functions $\hat{Z}$.

If we take $\psi_1(t) = \lambda t$ for all $t \geq 0$, $\lambda \in [0,1)$ and $\psi_2(t) = t$ then we obtain the following function $\eta_2(t,s) = \lambda s - t$ for all $t, s \in \mathbb{R}_+$ is in the class of functions $\hat{Z}$. If $s \leq t$ then $\eta_2(t,s) < 0$.

**Definition 5.** *Let* **F** *be the class of all functions* $G : \mathbb{R}_+ \times \mathbb{R}_+ \to \mathbb{R}_+$ *satisfying the following conditions:*

*(1)* $\max\{a,b\} \leq G(a,b)$ *for* $a, b \geq 0$.

*(2)* *G is continuous and nondecreasing.*

*(3)* $G(a+b, c+d) \leq G(a,c) + G(b,d)$.

For example $G(a,b) = a + b$.

## 2. Main Result

**Theorem 3.** *Let C be a nonempty, bounded, closed and convex subset of a Banach space* $\bar{E}$. *Also* $T : C \to C$ *is continuous and* $\phi : \mathbb{R}_+ \to \mathbb{R}_+$ *is continuous and nondecreasing functions. Suppose that if for any* $0 < a < b < \infty$ *there exists* $0 < \gamma(a,b) < 1$ *such that for all* $X \subseteq C$,

$$a \leq G(\mu(X), \phi(\mu(X))) \leq b \implies \eta\{G(\mu(TX), \phi(\mu(TX))), \gamma(a,b)G(\mu(X), \phi(\mu(X)))\} \geq 0,$$

*where* $\mu$ *is an arbitrary measure of noncompactness and* $\eta \in \hat{Z}$ *and* $G \in$ **F**. *Then T has at least one fixed point in C.*

**Proof.** Let us construct a sequence $(C_n)$ such that $C_0 = C$ and $C_{n+1} = \text{Conv}(TC_n)$ for $n \geq 0$. We observe that $TC_0 = TC \subseteq C = C_0$, $C_1 = \text{Conv}(TC_0) \subseteq C = C_0$, therefore by continuing this process, we have $C_0 \supseteq C_1 \supseteq C_2 \supseteq \ldots \supseteq C_n \supseteq C_{n+1} \supseteq \ldots$.

If there exists a natural number $m$ such that $\mu(C_m) = 0$ then $C_m$ is compact. By Schauder's fixed point theorem we conclude that $T$ has a fixed point.

So we assume that $\mu(C_n) > 0$ for some $n \geq 0$ i.e., $G(\mu(C_n), \phi(\mu(C_n))) > 0$ for all $n \geq 0$.

Let $X = C_n$ for some $n \in \mathbb{N}$.

For $a \leq G(\mu(C_n), \phi(\mu(C_n))) \leq b$ gives

$$\begin{aligned}
0 &\leq \eta\{G(\mu(TC_n), \phi(\mu(TC_n))), \gamma(a,b)G(\mu(C_n), \phi(\mu(C_n)))\} \\
&= \eta\{G(\mu(\text{Conv}TC_n), \phi(\mu(\text{Conv}TC_n))), \gamma(a,b)G(\mu(C_n), \phi(\mu(C_n)))\} \\
&= \eta\{G(\mu(C_{n+1}), \phi(\mu(C_{n+1}))), \gamma(a,b)G(\mu(C_n), \phi(\mu(C_n)))\} \\
&< \gamma(a,b)G(\mu(C_n), \phi(\mu(C_n))) - G(\mu(C_{n+1}), \phi(\mu(C_{n+1})))
\end{aligned}$$

i.e.,

$$\gamma(a,b) > \frac{G(\mu(C_{n+1}), \phi(\mu(C_{n+1})))}{G(\mu(C_n), \phi(\mu(C_n)))}.$$

If $G(\mu(C_{n+1}), \phi(\mu(C_{n+1}))) \geq G(\mu(C_n), \phi(\mu(C_n)))$ then $\gamma(a,b) > 1$ which is a contradiction hence $G(\mu(C_{n+1}), \phi(\mu(C_{n+1}))) < G(\mu(C_n), \phi(\mu(C_n)))$ for all $n \in \mathbb{N}$. Hence $\{G(\mu(C_n), \phi(\mu(C_n)))\}$ is a nonnegative decreasing sequence so there exists $\alpha \geq 0$ such that $\lim_{n \to \infty} G(\mu(C_n), \phi(\mu(C_n))) = \alpha$. Suppose $\alpha > 0$. Then, $0 < \alpha = a \leq G(\mu(C_n), \phi(\mu(C_n))) \leq G(\mu(C_0), \phi(\mu(C_0))) = b$ for all $n \geq 0$.

Again, we have for $X = C_n$ there exists $0 < \gamma(a,b) < 1$ such that

$$\begin{aligned}
0 &\leq \eta\{G(\mu(TC_n), \phi(\mu(TC_n))), \gamma(a,b)G(\mu(C_n), \phi(\mu(C_n)))\} \\
&= \eta\{G(\mu(C_{n+1}), \phi(\mu(C_{n+1}))), \gamma(a,b)G(\mu(C_n), \phi(\mu(C_n)))\}.
\end{aligned}$$

Let $G(\mu(C_{n+1}), \phi(\mu(C_{n+1}))) = t_n$, $\gamma(a,b)G(\mu(C_n), \phi(\mu(C_n))) = s_n$.

Since $t_n < s_n$ for all $n \geq 0$ and $\lim_{n \to \infty} t_n = \alpha$, $\lim_{n \to \infty} s_n = \gamma(a,b)\alpha$ therefore

$$\limsup_{n \to \infty} \eta \left\{ G(\mu(TC_n), \phi(\mu(TC_n))), \gamma(a,b) G(\mu(C_n), \phi(\mu(C_n))) \right\} < \gamma(a,b)\alpha - \alpha < 0$$

which is a contradiction. Thus we conclude $\alpha = 0$ i.e., $\lim_{n \to \infty} G(\mu(C_n), \phi(\mu(C_n))) = 0$. Hence we get $\lim_{n \to \infty} \mu(C_n) = 0$ and $\lim_{n \to \infty} \phi(\mu(C_n)) = 0$.

Since $C_n \supseteq C_{n+1}$ in the view of Definition 1, we conclude that $C_\infty = \bigcap_{n=1}^{\infty} C_n$ is nonempty, closed and convex subset of $C$ and $C_\infty$ is invariant under $T$. Thus Schauder's theorem implies that $T$ has a fixed point in $C_\infty \subseteq C$. This completes the proof. $\square$

**Theorem 4.** *Let $C$ be a nonempty, bounded, closed and convex subset of a Banach space $\bar{E}$. Also $T : C \to C$ is continuous and $\phi : \mathbb{R}_+ \to \mathbb{R}_+$ is continuous and nondecreasing functions. Suppose that if for any $0 < a < b < \infty$ there exists $0 < \gamma(a,b) < 1$ such that for all $X \subseteq C$,*

$$a \leq \mu(X) + \phi(\mu(X)) \leq b \implies \eta \left\{ \mu(TX) + \phi(\mu(TX)), \gamma(a,b)(\mu(X) + \phi(\mu(X))) \right\} \geq 0,$$

*where $\mu$ is an arbitrary measure of noncompactness and $\eta \in \hat{Z}$. Then $T$ has at least one fixed point in $C$.*

**Proof.** The result follows by taking $G(a,b) = a + b$ in Theorem 3. $\square$

**Theorem 5.** *Let $C$ be a nonempty, bounded, closed and convex subset of a Banach space $\bar{E}$ and $T : C \to C$ is a continuous function. Suppose that if for any $0 < a < b < \infty$ then there exists $0 < \gamma(a,b) < 1$ such that for all $X \subseteq C$,*

$$a \leq \mu(X)) \leq b \implies \eta \left\{ \mu(TX), \gamma(a,b)\mu(X) \right\} \geq 0$$

*where $\mu$ is an arbitrary measure of noncompactness and $\eta \in \hat{Z}$. Then $T$ has at least one fixed point in $C$.*

**Proof.** The result follows by taking $G(a,b) = a + b$ and $\phi \equiv 0$ in Theorem 3. $\square$

**Theorem 6.** *Let $C$ be a nonempty, bounded, closed and convex subset of a Banach space $\bar{E}$ and $T : C \to C$ is a continuous function. Suppose $\psi_1$ and $\psi_2$ be two altering distance functions such that $\psi_1(t) < t \leq \psi_2(t)$ for all $t > 0$ and a constant $0 < \gamma < 1$ such that for all $X \subseteq C$, and $a \leq \mu(X) \leq b$ we have $\psi_2(\mu(T(X))) \leq \psi_1(\gamma\mu(X))$ where $\mu$ is an arbitrary measure of noncompactness. Then $T$ has at least one fixed point in $C$.*

**Proof.** The result follows by taking $\eta(t,s) = \psi_1(s) - \psi_2(t)$ for all $t, s \geq 0$ in Theorem 5. $\square$

**Theorem 7.** *Let $C$ be a nonempty, bounded, closed and convex subset of a Banach space $\bar{E}$ and $T : C \to C$ is a continuous function. Suppose for any $0 < a < b < \infty$ there exists $0 < \gamma(a,b) < 1$ such that for all $X \subseteq C$, and $a \leq \mu(X) \leq b$ we have $\mu(T(X)) \leq \gamma(a,b)\mu(X)$, where $\mu$ is an arbitrary measure of noncompactness. Then $T$ has at least one fixed point in $C$.*

**Proof.** The result follows by taking $\eta(t,s) = \lambda s - t$ for all $t, s \geq 0$ and $\gamma(a,b) = \lambda\hat{\gamma}(a,b)$ in Theorem 6 where $\lambda \in [0,1)$ and $0 < \hat{\gamma}(a,b) < 1$. $\square$

## 3. Application

In this article, we shall work in the space $E = C([0,1] \times [0,1])$ which consists of the set of real continuous on $[0,1] \times [0,1]$. The space $E$ is equipped with the norm

$$\| x \| = \sup \left\{ |x(t,s)| : t,s \geq 0 \right\}, \ x \in E.$$

The space $E$ has the Banach algebra structure.

Let $X$ be a fixed nonempty and bounded subset of the space $E = C([0,1] \times [0,1])$ and for $x \in X$ and $\epsilon > 0$, denote by $\omega(x, \epsilon)$ the modulus of the continuity function $x$ i.e.,

$$\omega(x, \epsilon) = \sup \left\{ |x(t,s) - x(u,v)| : t, s, u, v \in [0,1], |t - u| \leq \epsilon, |s - v| \leq \epsilon \right\}.$$

Further we define

$$\omega(X, \epsilon) = \sup \left\{ \omega(x, \epsilon) : x \in X \right\}.$$

$$\omega_0(X) = \lim_{\epsilon \to 0} \omega(X, \epsilon).$$

Similar to [5] it can be shown that the function $\omega_0$ is a measure of non-compactness in the space $C([0,1] \times [0,1])$.

In this part we are going to study the existence of the solution of the following integral equation

$$x(t,s) = G(t,s) + F\left(t, s, x(t,s), \int_0^t \int_0^s u(t,s,v,w,x(v,w))dvdw\right), \ t, s \in [0,1] = I. \tag{1}$$

We consider the following assumptions

(1) The function $G : I \times I \to \mathbb{R}$ is continuous and nondecreasing. Also $B = \sup \left\{ |G(t,s)| : t, s \in I \right\}$.

(2) Let $u : I \times I \times I \times I \times \mathbb{R} \to \mathbb{R}$ is continuous function such that $u : I \times I \times I \times I \times \mathbb{R}_+ \to \mathbb{R}_+$ and for arbitrary fixed $v, w \in I$ and $x \in \mathbb{R}_+$ we have $u(t,s,v,w,x)$ is nondecreasing. Also, $|u(t,s,v,w,x)| \leq L|x|$ for $t, s, v, w \in I$; $x \in \mathbb{R}$ and $L \geq 0$.

(3) The function $F : I \times I \times \mathbb{R} \times \mathbb{R} \to \mathbb{R}$ is continuous such that there exists $K \in [0,1)$ satisfying

$$|F(t,s,x,y) - F(t,s,\bar{x},\bar{y})| \leq K|x - \bar{x}| + |y - \bar{y}|$$

and $M = \sup \left\{ |F(t,s,0,0)| : t, s \in I \right\}$.

(4) There exists $r > 0$ such that $B + M + (K + L)r < r$.

Let the closed ball with center 0 and radius $r$ be denoted by $B_r = \left\{ x \in C(I \times I) : \| x \| \leq r \right\}$.

**Theorem 8.** *Under the hypothesis (1)–(4), Equation (1) has at least one solution in $C(I \times I)$, where $I = [0,1]$.*

**Proof.** Let us consider the operators $\hat{F}$ and $\hat{T}$ defined on $C(I \times I)$ as follows

$$(\hat{F}x)(t,s) = F\left(t, s, x(t,s), \int_0^t \int_0^s u(t,s,v,w,x(v,w))dvdw\right)$$

and

$$(\hat{T}x)(t,s) = G(t,s) + (\hat{F}x)(t,s)$$

where $t, s \in I$.

From assumptions (1) to (3) we infer $\hat{T}x$ is continuous on $I \times I$ for $x \in C(I \times I)$. Thus $\hat{T}$ maps $C(I \times I)$ into itself. Also for $t, s \in I$ we get

$$|(\hat{F}x)(t,s)|$$

$$\leq \left| F\left(t,s,x(t,s),\int_0^t \int_0^s u(t,s,v,w,x(v,w))dvdw\right) - F(t,s,0,0) \right| + |F(t,s,0,0)|$$

$$\leq K|x(t,s)| + \left| \int_0^t \int_0^s u(t,s,v,w,x(v,w))dvdw \right| + M$$

$$\leq K|x(t,s)| + \int_0^t \int_0^s L|x(v,w)|\,dvdw + M$$

$$\leq K|x(t,s)| + L\parallel x \parallel + M$$

$$\leq (K+L)\parallel x \parallel + M.$$

Then we have

$$|(\hat{T}x)(t,s)|$$
$$\leq |G(t,s)| + |(\hat{F}x)(t,s)|$$
$$\leq B + (K+L)\parallel x \parallel + M.$$

Thus if $\parallel x \parallel \leq r$, we have $|(\hat{T}x)(t,s)| \leq B + (K+L)r + M \leq r$ i.e., $\parallel \hat{T}x \parallel \leq r$.

Therefore the operator $\hat{T}$ maps $B_r$ into itself.

Next we have to prove that $\hat{T}$ is continuous on $B_r$. Let $\{x_n\}$ be a sequence in $B_r$ such that $x_n \to x$. For every $t,s \in I$, we have

$$|(\hat{F}x_n)(t,s) - (\hat{F}x)(t,s)|$$
$$= \left| F\left(t,s,x_n(t,s),\int_0^t \int_0^s u(t,s,v,w,x_n(v,w))dvdw\right) - F\left(t,s,x(t,s),\int_0^t \int_0^s u(t,s,v,w,x(v,w))dvdw\right) \right|$$
$$\leq K|x_n(t,s) - x(t,s)| + \int_0^t \int_0^s |u(t,s,v,w,x_n(v,w)) - u(t,s,v,w,x(v,w))|\,dvdw$$
$$\leq K\parallel x_n - x \parallel + U_r(\epsilon),$$

where $\epsilon > 0$ and

$$U_r(\epsilon) = \sup\left\{|u(t,s,v,w,x) - u(t,s,v,w,\bar{x})| : t,s,v,w \in I; x,\bar{x} \in [-r,r]; \parallel x - \bar{x} \parallel < \epsilon\right\}.$$

As

$$|(\hat{T}x_n)(t,s) - (\hat{T}x)(t,s)|$$
$$\leq K\parallel x_n - x \parallel + U_r(\epsilon)$$

It follows $\parallel \hat{T}x_n - \hat{T}x \parallel \leq K\parallel x_n - x \parallel + U_r(\epsilon)$.

As $\epsilon \to 0$ we get $U_r(\epsilon) \to 0$ because $u$ is uniformly continuous on $I \times I \times I \times I \times [-r,r]$. Thus $\parallel \hat{T}x_n - \hat{T}x \parallel \to 0$. Hence $\hat{T}$ is continuous on $B_r$.

Let us consider an nonempty subset $X$ of $B_r$ and $x \in X$ then for a fixed $\epsilon > 0$ and $t_1,t_2,s_1,s_2 \in I$ such that $t_1 \leq t_2$, $s_1 \leq s_2$, $|t_1 - t_2| \leq \epsilon$, $|s_1 - s_2| \leq \epsilon$.

Then we get

$$
\left| (\hat{F}x)(t_2, s_2) - (\hat{F}x)(t_1, s_1) \right|
$$

$$
\leq \left| F\left( t_2, s_2, x(t_2, s_2), \int_0^{t_2} \int_0^{s_2} u(t_2, s_2, v, w, x(v, w)) dv dw \right) \right.
$$

$$
\left. - F\left( t_2, s_2, x(t_1, s_1), \int_0^{t_2} \int_0^{s_2} u(t_2, s_2, v, w, x(v, w)) dv dw \right) \right|
$$

$$
+ \left| F\left( t_2, s_2, x(t_1, s_1), \int_0^{t_2} \int_0^{s_2} u(t_2, s_2, v, w, x(v, w)) dv dw \right) \right.
$$

$$
\left. - F\left( t_1, s_1, x(t_1, s_1), \int_0^{t_2} \int_0^{s_2} u(t_2, s_2, v, w, x(v, w)) dv dw \right) \right|
$$

$$
+ \left| F\left( t_1, s_1, x(t_1, s_1), \int_0^{t_2} \int_0^{s_2} u(t_2, s_2, v, w, x(v, w)) dv dw \right) \right.
$$

$$
\left. - F\left( t_1, s_1, x(t_1, s_1), \int_0^{t_2} \int_0^{s_2} u(t_1, s_1, v, w, x(v, w)) dv dw \right) \right|
$$

$$
+ \left| F\left( t_1, s_1, x(t_1, s_1), \int_0^{t_2} \int_0^{s_2} u(t_1, s_1, v, w, x(v, w)) dv dw \right) \right.
$$

$$
\left. - F\left( t_1, s_1, x(t_1, s_1), \int_0^{t_1} \int_0^{s_1} u(t_1, s_1, v, w, x(v, w)) dv dw \right) \right|
$$

$$
\leq K \left| x(t_2, s_2) - x(t_1, s_1) \right| + \omega(F, \epsilon)
$$

$$
+ \int_0^{t_2} \int_0^{s_2} \left| u(t_2, s_2, v, w, x(v, w)) - u(t_1, s_1, v, w, x(v, w)) \right| dv dw
$$

$$
+ \int_{t_1}^{t_2} \int_{s_1}^{s_2} \left| u(t_1, s_1, v, w, x(v, w)) \right| dv dw
$$

$$
\leq K \left| x(t_2, s_2) - x(t_1, s_1) \right| + \omega(F, \epsilon) + \int_0^{t_2} \int_0^{s_2} \omega(u, \epsilon) dv dw + \int_{t_1}^{t_2} \int_{s_1}^{s_2} \bar{U} dv dw
$$

$$
\leq K \left| x(t_2, s_2) - x(t_1, s_1) \right| + \omega(F, \epsilon) + \omega(u, \epsilon) + \bar{U} \epsilon^2,
$$

where

$$
\omega(u, \epsilon) = \sup \left\{ \begin{array}{c} \left| u(t_2, s_2, v, w, x) - u(t_1, s_1, v, w, x) \right| : t_1, t_2, s_1, s_2, v, w \in I, \\ |t_2 - t_1| \leq \epsilon, |s_2 - s_1| \leq \epsilon, x \in [-r, r] \end{array} \right\},
$$

$$
\bar{U} = \sup \left\{ \left| u(t, s, v, w, x) \right| : t, s, v, w \in I, x \in [-r, r] \right\}
$$

and

$$
\omega(F, \epsilon) = \sup \left\{ \begin{array}{c} \left| F(t, s, x, y) - F(t_1, s_1, x, y) \right| : t, t_1, s, s_1 \in I, \\ |t - t_1| \leq \epsilon, |s - s_1| \leq \epsilon, x \in [-r, r], y \in [-\bar{U}, \bar{U}] \end{array} \right\}.
$$

Hence

$$
\left| (\hat{T}x)(t_2, s_2) - (\hat{T}x)(t_1, s_1) \right|
$$

$$
\leq \left| G(t_2, s_2) - G(t_1, s_1) \right| + \left| (\hat{F}x)(t_2, s_2) - (\hat{F}x)(t_1, s_1) \right|
$$

$$
\leq \omega(G, \epsilon) + K \left| x(t_2, s_2) - x(t_1, s_1) \right| + \omega(F, \epsilon) + \omega(u, \epsilon) + \bar{U} \epsilon^2,
$$

where

$$
\omega(G, \epsilon) = \sup \left\{ \begin{array}{c} \left| G(t_2, s_2) - G(t_1, s_1) \right| : t_1, t_2, s_1, s_2 \in I, \\ |t_2 - t_1| \leq \epsilon, |s_2 - s_1| \leq \epsilon \end{array} \right\}.
$$

Now taking the supremum on $x$, we get

$$\omega(\hat{T}X, \epsilon) \leq \omega(G, \epsilon) + K\omega(X, \epsilon) + \omega(F, \epsilon) + \omega(u, \epsilon) + \bar{U}\epsilon^2.$$

Since $G$, $F$ and $u$ are uniformly continuous on $I \times I$, $I \times I \times [-r, r] \times [-\bar{U}, \bar{U}]$ and $I \times I \times I \times [-r, r]$ respectively therefore, we get, $\omega(G, \epsilon) \to 0$, $\omega(F, \epsilon) \to 0$ and $\omega(u, \epsilon) \to 0$ as $\epsilon \to 0$. Thus we obtain

$$\omega_0(\hat{T}X) \leq K\omega_0(X).$$

This implies $\hat{T}$ is a contraction operator on $B_r$ with respect to $\omega_0$. Thus by Theorem 7, we have $\hat{T}$ has at least one fixed point in $B_r$. Hence Equation (1) has at least one solution in $B_r \subset C(I \times I)$. This completes the proof. $\square$

**Example 1.** *Consider the following equation*

$$x(t, s) = \frac{ts}{1 + ts} + \frac{t^2 s^2 x(t, s)}{4(1 + t^2 s^2)} + \frac{ts}{4} \int_0^t \int_0^s vw \sin(x(v, w)) dv dw \tag{2}$$

*for $t, s \in [0, 1] = I$.*
*Here we have*

$$G(t, s) = \frac{ts}{1 + ts},$$

$$F(t, s, x, y) = \frac{t^2 s^2 x}{4(1 + t^2 s^2)} + y,$$

$$u(t, s, v, w, x) = \frac{tsvw \sin x}{4}.$$

*It can be easily seen that $G$, $u$ are continuous functions on $I \times I$ and $I \times I \times I \times I \times \mathbb{R}$, respectively. The function $u$ is nondecreasing and*

$$|u(t, s, v, w, x)| \leq \frac{1}{4} |x|.$$

*Also we have $B = 1$ and $L = \frac{1}{4}$.*
*The function $F$ is continuous on $I \times I \times \mathbb{R} \times \mathbb{R}$ and*

$$|F(t, s, x, y) - F(t, s, \bar{x}, \bar{y})|$$
$$= \left| \frac{t^2 s^2 x}{4(1 + t^2 s^2)} + y - \frac{t^2 s^2 \bar{x}}{4(1 + t^2 s^2)} - \bar{y} \right|$$
$$\leq \frac{t^2 s^2}{4(1 + t^2 s^2)} |x - \bar{x}| + |y - \bar{y}|$$
$$\leq \frac{1}{4} |x - \bar{x}| + |y - \bar{y}|.$$

*Here $K = \frac{1}{4}$ and $M = 0$.*
*The inequality in the assumption (4) has the following form*

$$1 + \frac{r}{2} < r.$$

*For $r = 3$ we observe that all the assumption from (1)–(4) of Theorem 8 are satisfied. Thus applying the Theorem 8 we conclude that the Equation (2) has at least one solution in $C(I \times I)$.*

**Funding:** This project was supported by Theoretical and Computational Science (TaCS) Center under Computational and Applied Science for Smart Innovation research Cluster (CLASSIC), Faculty of Science, KMUTT.

**Acknowledgments:** The authors acknowledge the financial support provided by King Mongkut's University of Technology Thonburi through the "KMUTT 55th Anniversary Commemorative Fund".

**Conflicts of Interest:** The authors declare that they have no competing interests.

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
