# Peer review of "Some New Generalization of Darbo’s Fixed Point Theorem and Its Application on Integral Equations"

_mathematics, doi:10.3390/math7030214_

Round 1

Reviewer 1 Report

1-     Some typos do appear. A carefully reading is necessary.

2-     Why are simulation functions called thus? Why the word simulation?

3-     Why are altering distance functions called thus? Why the word altering?

4-    Line 65: So defined the values of eta may be negative!

5-    Line 81: Does the equation occurs for one function G (which ones ?), all functions G

6-    Line 112: Does the equation occurs for one function eta (which ones ?), all functions eta

Author Response

Manuscript ID: mathematics-424143 

Title: Some new generalization of Darbo's fixed point theorem and its application on integral equations
Authors: Anupam Das, Bipan Hazarika, Poom Kumam 

We have revised the article according to the referees comments.

Response to referees comments:

1.     We replaced the word simulation function by control function in abstract because by mistake we wrote simulation function

2.     We correct all the typos

3.     We add the reference of Khan et al. [17] where the authors used “control function” which they called “altering distance function “

4.     We add an example to verify to checked the negativity of the function $\eta$ in line 1-2 from top in page 3

5.     In Theorem 2.1 mentioned that:  for all function $G\in \mathbf{F}$

6.     In Theorems 2.2 and 2.3 we mentioned that: for all function $\eta\in \hat{Z}$

7.     We add the references [7,8,13,14,15,17]

Authors.

Reviewer 2 Report

The work seems correct and well organized. I think the work is interesting for those working in this field. I recommend careful reading to correct some writing errors.

Author Response

Manuscript ID: mathematics-424143 

Title: Some new generalization of Darbo's fixed point theorem and its application on integral equations
Authors: Anupam Das, Bipan Hazarika, Poom Kumam 

We have revised the article according to the referees comments.

Response to referees comments:

1.     We replaced the word simulation function by control function in abstract because by mistake we wrote simulation function

2.     We correct all the typos

3.     We add the reference of Khan et al. [17] where the authors used “control function” which they called “altering distance function “

4.     We add an example to verify to checked the negativity of the function $\eta$ in line 1-2 from top in page 3

5.     In Theorem 2.1 mentioned that:  for all function $G\in \mathbf{F}$

6.     In Theorems 2.2 and 2.3 we mentioned that: for all function $\eta\in \hat{Z}$

7.     We add the references [7,8,13,14,15,17]

Best regards,

Poom Kumam